# What Is Missing: Interpretable Ratings for Large Language Model Outputs

## Abstract

Current Large Language Model (LLM) preference learning methods such as Proximal Policy Optimization and Direct Preference Optimization rely on direct rankings or numerical ratings of model outputs as a way to learn human preferences. These rankings are subjective, and a single numerical rating chosen directly by a judge is a poor metric to quantify a complex system such as human language. This paper introduces the *What Is Missing* (WIM) rating system to create better rankings for preference learning methods. WIM is a straightforward method that can be integrated into existing training pipelines, combined with other rating techniques, and used as the input to any preference learning method without changes. To create a WIM rating, natural language feedback for a model output is given by a human or LLM judge. Both the output and the feedback are passed through a sentence embedding model and the cosine similarity between the high dimensional vectors is calculated. Theoretical benefits in the distribution of WIM ratings, compared to numerical ratings, translate into lower loss throughout training, better reward advantage scaling, and better performance in a trained task. Importantly, WIM is interpretable as the reason for the chosen ranking can be discovered easily. WIM provides an alternate way to think about preference learning by shifting the focus away from the algorithms themselves and onto the improvement of the preference data generation pipeline.

## 1 Introduction

The creation of the Large Language Model (LLM) has changed what humans can do with a computer Brown et al. (2020). To achieve these technological breakthroughs, a large corpus of data and training resources is required Kandpal & Raffel (2025). The training time and resources are split between two distinct phases: pre-training and post-training. An LLM that has been pre-trained is an excellent next word prediction machine and has some ability to perform instruction-following tasks Radford et al. (2019).

The second phase, post-training, can be broken into two categories: the Supervised Fine-Tuning (SFT) phase and the preference learning phase Fernando et al. (2025). The SFT phase can train an LLM to produce specific outputs by minimizing a cross entropy loss on an instruction-following dataset. The preference learning phase aims to improve the usefulness of the LLMs by tuning the model to human preferences Ouyang et al. (2022). As human preferences cannot be directly calculated, the preference learning phase requires the use of a reward model and reinforcement learning (RL) instead of a direct loss function Ouyang et al. (2022).

Expanding the post-training tool set will allow researchers to better prevent misalignment. Misalignment is described as the difference between human goals and the objectives of the LLM Christiano et al. (2017). Misalignment is an ever-increasing problem as model performance continues to improve and is crucial as the newest models have warranted new safety protections, such as Anthropic's AI Safety Level 3 (ASL-3) designation for Claude Opus 4 Anthropic (2025). If models acquire superhuman intelligence, they must be aligned to human goals and values or there is a potential for catastrophic damage to human civilization Carlsmith (2024).

One of the primary tools for addressing misalignment is preference learning, where the training loop revolves around ranking model outputs and optimizing the model on that ranking. Historically, the ranking system was decided using direct rankings as human evaluators would directly rank

completions using their own preferences Ouyang et al. (2022). Rankings are subjective, relying on heuristic evaluations and user preferences rather than clear performance metrics Kumar et al. (2025). A method to understand why every ranking was chosen is impossible as different judges will not always create the same ranking. An improvement on the ranking system is to score each output using a numerical rating system such as a scale from 1-10, as seen in Lee et al. (2024a). This allows outputs in a ranking to be compared with each other and can demonstrate how much better or worse responses are compared with each other.

Fundamentally, a numerical rating system has the same shortfalls as a direct ranking system because it is difficult to distill the worth of an answer into a single number. Outputs with the same rating can differ significantly. A numerical rating system is a discrete set and is a poor tool to quantify a complex system such as human language. Experiments in Section 3.2 empirically demonstrate that numerical ratings can produce the same rating in a pairwise comparison, preventing the generation of a learning signal. As the use of synthetic data increases, other LLM systems have taken the role of the judge Lee et al. (2024b). LLM feedback systems still rely on the same methods to rank and therefore, post-train models.

Overall, shortcomings of the existing ranking and rating systems can be classified into the following categories: the low interpretability of the ratings and the same ratings preventing the creation of a learning signal. This paper introduces *What Is Missing* (WIM) feedback scoring as an alternative to traditional numerical ratings or direct rankings. WIM provides natural language feedback, making the rating directly interpretable. The produced rating distribution are discrete samples of a continuous distribution and therefore, repeated ratings are much less frequent. Both of these improvements position WIM as a solution to increase preference learning performance while simultaneously providing interpretable ratings.

## 2 THE PROPOSED WIM METHOD

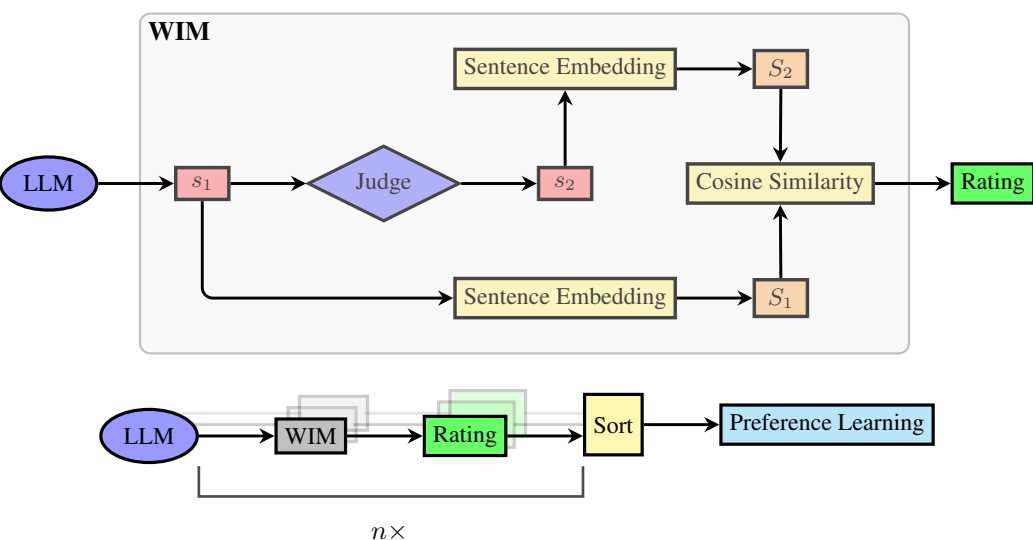

Figure 1: Flowchart of the WIM method. An LLM produces a natural language output $s_1$. $s_1$ is then evaluated by a human or an LLM judge. The judge's goal is to produce $s_2$, a response containing what is missing in $s_1$. Both $s_1$ and $s_2$ are passed through a sentence embedding model to produce high dimensional vectors $S_1$ and $S_2$. The similarity of $S_1$ and $S_2$ is calculated using cosine similarity and the resulting similarity score is the WIM rating. A higher similarity between $S_1$ and $S_2$ implies that there is less missing from the LLM's output. $n$ model outputs are rated by the WIM method and then sorted to produce a ranking. The ranking of the outputs is then passed to a preference learning algorithm.

The process of creating *What Is Missing* (WIM) feedback scoring is demonstrated in Figure 1. In WIM, a human or LLM judge produces a natural-language description of what was missing in the

model's output. For example, if the model forgets to mention a keypoint in its argument or forgot some functionality when performing a coding task.

Conceptually, this process is adversarial: the model aims to include all relevant information, while the judge identifies missing elements. This dynamic is similar to the discriminator in a Generative Adversarial Network, although here the goal is to surface missing content rather than distinguish between real and generated examples Goodfellow et al. (2014).

The scoring procedure works as follows:

1. The base model output ($s_1$) and the WIM response ($s_2$) are each passed through a sentence embedding model, producing high-dimensional vector representations $S_1$ and $S_2$ Reimers & Gurevych (2019). These embeddings capture semantic properties of each text.

2. Cosine similarity is computed between $S_1$ and $S_2$ to quantify semantic overlap Mikolov et al. (2013).

3. The resulting score, in $[-1, 1]$, serves as the feedback rating for the base model's output. If no WIM feedback is provided (i.e., nothing was missing), a perfect score of 1 is assigned as a design choice.

4. Once a WIM score for all outputs being compared has been computed, the scores are ranked from highest to lowest. The ranking can then be used as input to any preference learning algorithm such as Proximal Policy Optimization (PPO) or Direct Preference Optimization (DPO) Schulman et al. (2017) Rafailov et al. (2023).

### 2.1 MATHEMATICAL EXPLANATION

Referring again to Figure 1, let the model's generated output be a sequence of $n$ tokens:

$$s_1 = [w_1, w_2, \ldots, w_n], \tag{1}$$

where $w_i$ is the $i$-th token.

The WIM response is a sequence of $m$ tokens describing what $s_1$ omitted:

$$s_2 = [w'_1, w'_2, \ldots, w'_m]. \tag{2}$$

A sentence embedding function $f_{\text{embedding}}$ maps each sequence into a vector in $\mathbb{R}^d$:

$$S_1 = f_{\text{embedding}}(s_1) \in \mathbb{R}^d, \quad S_2 = f_{\text{embedding}}(s_2) \in \mathbb{R}^d. \tag{3}$$

The WIM score is computed as the cosine similarity between these vectors:

$$\text{WIM} = \frac{S_1 \cdot S_2}{\|S_1\| \, \|S_2\|}. \tag{4}$$

A higher WIM score indicates that the model's output and the WIM feedback are more semantically similar, suggesting less *missingness*.

### 2.2 MISSINGNESS

The WIM vector, $S_2$ can be decomposed as shown in Equation 5:

$$S_2 = \underbrace{proj_{\mathbf{S_1}} \mathbf{S_2}}_{\text{parallel feedback}} + \underbrace{(S_2 - proj_{\mathbf{S_1}} \mathbf{S_2})}_{\text{orthogonal feedback}} = S_2^{\parallel} + S_2^{\perp} \tag{5}$$

*Missingness* represents the missing content in the model output and can be thought of as the *orthogonal feedback vector*, $S_2^{\perp}$. As this vector grows in relation to the *parallel feedback vector*, $S_2^{\parallel}$, the amount of missing content in the model's response should grow proportionally. Orthogonality to $S_1$

implies $S_1^\top S_2^\perp = 0$ and is equivalent to having no common information in an embedding space. As $S_2^\perp$ grows in relation to $S_2^\parallel$, the angle between vectors $S_1$ and $S_2$ will also increase as the missingness vector's magnitude is given as $\|S_2^\perp\| = \|S_2\|\sqrt{1 - \cos^2\theta}$ (Appendix A.3). The angle in the $\|S_2^\perp\|$ means cosine similarity is a valid metric to measure missingness. The relationship between missingness and cosine similarity can be visualized on the 2D plane in Figure 2. Note the same interpretation can be taken even when $S_2^\parallel$ is antiparallel as that case would result in a negative WIM rating.

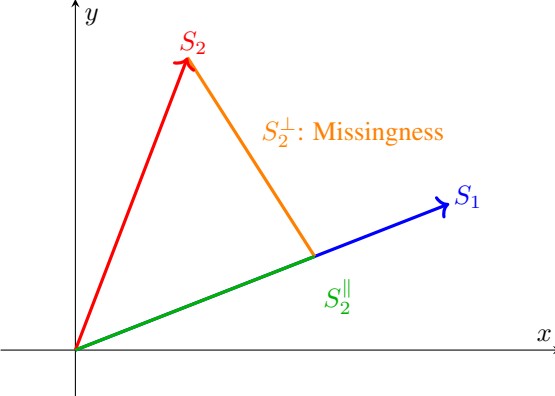

Figure 2: 2D visualization of missingness

### 2.3 RANKING USAGE

Online Direct Preference Optimization (ODPO) was chosen as the preference learning algorithm to optimize the model for the selected ranking Guo et al. (2024). ODPO was chosen because it does not require the training of a reward model and, since it is online, the model can be trained using LLM inference instead of creating a dataset of responses.

As WIM aims to improve how the ranking of model output is determined, ODPO is not required for this process. WIM is agnostic to the training process as it only aims to improve the rankings for any existing preference learning algorithm such as Proximal Policy Optimization (PPO) and Group Relative Policy Optimization (GRPO) Schulman et al. (2017) Shao et al. (2024). Being algorithm agnostic allows WIM to be directly implemented into existing training infrastructure, saving engineering costs and the time to launch. Other natural language feedback systems such as Text2Grad require a completely new training process including training a separate reward model Wang et al. (2025). Alternative training techniques with unique post-training approaches such as Constitutional AI, still contain a preference learning phase after a unique SFT phase Bai et al. (2022). As new preference learning methods are created, WIM will continue to be useful as long as preference learning is based on rankings.

### 2.4 SELF-JUDGING

There is no requirement that the WIM response is created by a human. A larger and more powerful model could be used to produce the WIM feedback. The model being trained can also be used as a self-judge to reflect on its own output Yuan et al. (2024). Self-judging can be thought of as a researcher reviewing the first draft of their paper and refining the ideas they have created.

This paper switched the context window of the LLM being trained as the critic's task is less difficult than the actor. Both the LLM's fixed reference model (*Fixed Judge*) and the model being actively post-trained (*Moving Judge*) were used as the judge. The self-critic paradigm simplified the training requirements while also using the special benefit of ODPO not needing a pre-trained reward model. Using the self-critic paradigm prevents the model from being limited by human capabilities or by the performance of the current state-of-the-art (STOA) model. The self-critic paradigm could be used to train an STOA model if the actor and critic improved in parallel.

## 3 THEORETICAL ANALYSIS

The WIM feedback system is interpretable as the score is derived from the feedback used to create the rating. Even if this method performed equally to numerical rating systems, it would be preferable because of the added interpretability. Outside of interperetability, theoretical benefits in rating distribution and the winning and losing rating difference were explored. Theoretical results were calculated using data collected during the rating process. A 1-10 rating scale was used following the LLM rating approach in Lee et al. (2024a). The numerical 1-10 rating was compared with WIM as both of these methods produce scalar ratings that can be compared, unlike a ranking where outputs are sorted directly.

The 1-10 rating scale was also chosen as it lies within the empirically optimal range for psychological measurements. Psychological measurements are relevant to this field because the creation of preference learning data requires measuring the judgments of humans or LLMs. Test-retest reliability decreases in scales with more than 10 categories Preston & Colman (2000) and the psychometric properties of the rating scale plateau at 7 categories Lozano et al. (2008). Increasing the numerical rating scale does not meaningfully increase information density and can reduce the consistency of ratings.

### 3.1 RATING DISTRIBUTION

The original Direct Preference Optimization (DPO) loss function is shown in Equation 6 Rafailov et al. (2023). The loss function takes a winning output, and a losing output denoted $y_w$ and $y_l$ respectively. The DPO update shown in Equation 7 demonstrates that the model weights are updated to increase the likelihood of the winning policy ($y_w$) and decrease the likelihood of the losing policy ($y_l$). To increase the performance of this learning algorithm, it is beneficial to have a clear differentiator between winning and losing outputs. First the distribution of the numerical rating system can be examined. Figure 3a is a histogram of the numerical rating system given on a scale of 1 to 10 (-1 to 1 used in training). The numerical rating system is discrete and heavily clustered around a score of 7 and 8.

Figure 3b shows the WIM rating system. The distribution resembles discrete samples of a continuous distribution. Figure 3b also demonstrates that WIM's distribution is closer to a normal distribution than the numerical system. Note that the WIM distribution is negatively skewed and the large amount of 10 ratings were produced when an answer is determined to have nothing missing.

$$\mathcal{L}_{\text{DPO}}(\pi_\theta; \pi_{\text{ref}}) = -\mathbb{E}_{(x,y_w,y_l)\sim\mathcal{D}} \left[ \log \sigma \left( \beta \log \frac{\pi_\theta(y_w \mid x)}{\pi_{\text{ref}}(y_w \mid x)} - \beta \log \frac{\pi_\theta(y_l \mid x)}{\pi_{\text{ref}}(y_l \mid x)} \right) \right]. \quad (6)$$

$$\nabla_\theta \mathcal{L}_{\text{DPO}}(\pi_\theta; \pi_{\text{ref}}) =$$
$$- \beta \mathbb{E}_{(x,y_w,y_l)\sim\mathcal{D}} \left[ \underbrace{\sigma(\hat{r}_\theta(x, y_l) - \hat{r}_\theta(x, y_w))}_{\text{higher weight when reward estimate is wrong}} \left[ \underbrace{\nabla_\theta \log \pi(y_w \mid x)}_{\text{increase likelihood of } y_w} - \underbrace{\nabla_\theta \log \pi(y_l \mid x)}_{\text{decrease likelihood of } y_l} \right] \right]$$
$$(7)$$

### 3.2 WINNING AND LOSING RATING DIFFERENCE

The real differentiator for if a model will learn properly is if there is a clear rating separation between winning and losing outputs. Referring back to Equation 7, the likelihood of the policy producing the winning response ($y_w$) is increased and the likelihood of the policy producing the losing response ($y_l$) is decreased. If there is no rating separation between $y_w$ and $y_l$, policy updates can be counterproductive since the true $y_w$ and $y_l$ could be mislabeled. As seen in Figure 3a, most ratings are 7, 8, or 9 and having three rating groups drastically increases equal ratings.

The rating separation can be measured by comparing the difference or delta between the winning and losing output ratings. The rating distribution is shown in Figure 4a for the numerical rating system and in Figure 4b for the WIM rating system. Since the numerical rating system is discrete

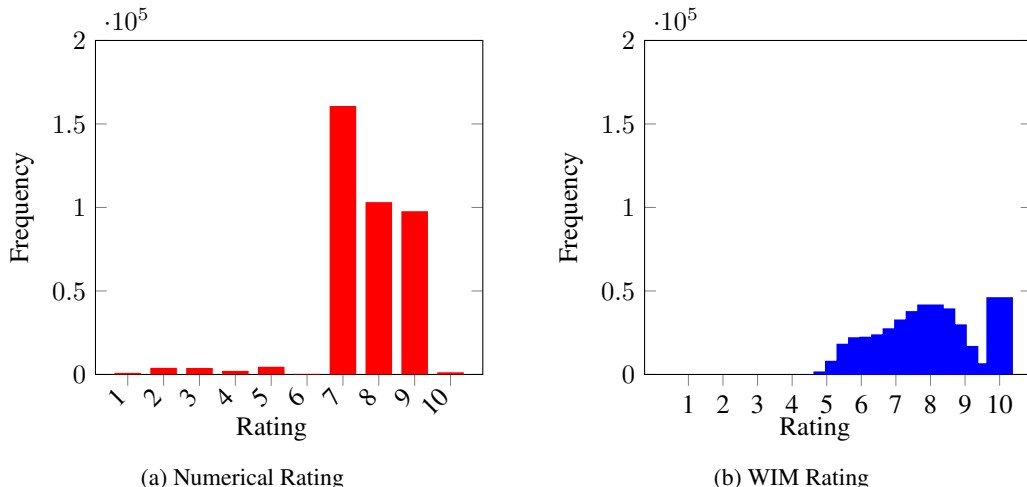

(a) Numerical Rating

(b) WIM Rating

Figure 3: Histogram of ratings from the Numerical rating system and the WIM rating system

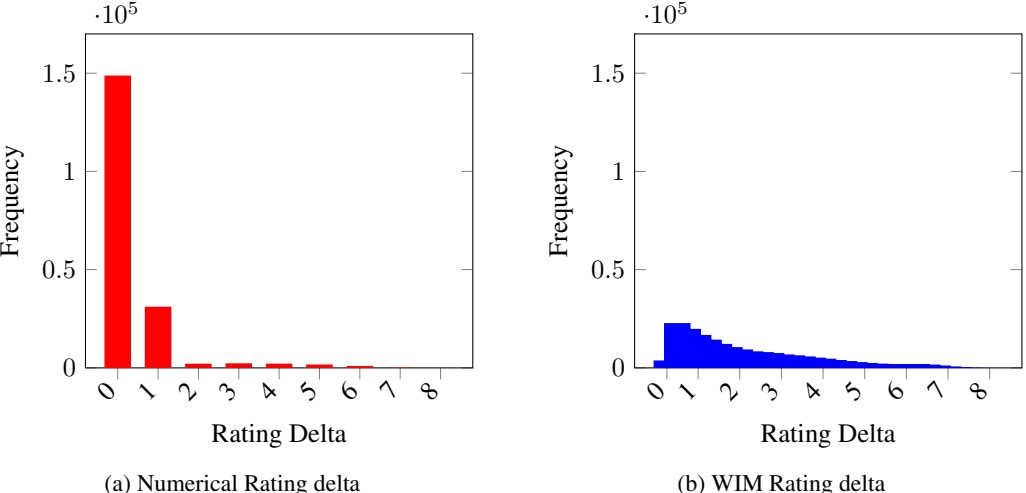

(a) Numerical Rating delta

(b) WIM Rating delta

Figure 4: Histogram of rating deltas from the Numerical Rating System and the WIM Rating System

and contains many duplicate ratings, there are many output pairs with no rating delta. No rating delta means that no learning signal can be produced from the judging of these responses. It is such a problem that 42.78% of output pairs were given the same rating in the numerical system compared with 2.00% in the WIM rating system. The average delta between answers for WIM is 47.82% higher (Table 1). The higher rating delta of WIM could lead to a clearer learning signal being generated by the WIM rating system. A learning method that used the ratings of the winning and losing responses in its loss function could further utilize the higher rating delta.

Table 1: Average rating delta per judging pair

| Method | Average Delta |
|---|---|
| Numerical | 0.928 |
| WIM | **1.396** |

### 3.3 Application to Other Learning Methods

As mentioned in Section 2.3, the theoretical benefits of WIM do not only apply to DPO and its variants. In a method such as PPO, the preference rankings are used to train the reward model that is then used to update the model's policy Schulman et al. (2017). The reward is created in a similar fashion to the loss function shown in Equation 8 Ziegler et al. (2020). This loss function is a cross-entropy loss that increases the reward value for the chosen model output. As WIM produces a larger delta between ratings or in this case variance of the rating distribution, loss updates for reward models would also be larger. Therefore, WIM has the potential to improve the training of reward models for other preference learning methods because of the beneficial ranking properties it exhibits.

$$\text{loss}(r) = \mathbb{E}_{(x,\{y_i\},b)\sim\mathcal{S}} \left[ \log \frac{e^{r(x,y_b)}}{\sum_i e^{r(x,y_i)}} \right] \tag{8}$$

## 4 Experiments

To test the performance difference between the numerical rating system and WIM, a Meta-Llama-3-8B-Instruct model was fine-tuned using the ultrafeedback-prompt dataset which includes general question-answer format questions Grattafiori et al. (2024) trl-lib (2024). The all-mpnet-base-v2 sentence embedding model was used as the embedding function for training Sentence-Transformers (2024). In each training run, the same judge created both the numerical rating and the WIM response. All other training parameters were the same to give a fair comparison.

Equation 9 shows how these ratings can be mixed and controlled using the hyperparameter zeta ($\zeta$). The ability for WIM to be mixed and combined with the numerical rating system or a binary rating system such as in Reinforcement Learning with Verifiable Rewards (RLVR) DeepSeek-AI et al. (2025), allows complex feedback to be distilled into a single scalar value to compare across outputs. This property is again useful because that allows for WIM to be integrated into existing training pipelines and with existing preference feedback methods. It is important to note, Equation 9 casts the 1 to 10 rating to a -1 to 1 rating, so the final rating is from -1 to 1. During training, the numerical rating system used a zeta of 0 and WIM used a zeta of 1. To rank model outputs for the DPO trainer, the highest rating was chosen as the best answer.

$$\text{reward} = (1 - \zeta)R + \zeta\,\text{WIM},$$
$$R = \frac{\text{rating} - \bar{r}}{\bar{r}},$$
$$\bar{r} = \frac{\max\_rating + \min\_rating}{2} \tag{9}$$

All models were trained using three Nvidia H100 80GB GPUs, a batch size of 64, and training for roughly 200 hours. Other configurations included the use of bfloat16 mixed precision training and the use of flash attention Dao et al. (2022). Memory saving techniques were used during the training process. LoRA was used to reduce the number of trainable parameters for the model Hu et al. (2022) and the 8-bit version of the Adam optimizer was used to reduce the storage of optimizer states Dettmers et al. (2023). Finally, a context switching sequence was developed for the judging and training LLM to stop the need for the initialization of another LLM judge.

### 4.1 Training Metrics

To confirm that the theoretical benefits of the WIM rating system translate to better performance, training metrics were tracked and analyzed. Specifically, the training loss, mean model entropy, and the chosen and rejected rewards. A Random Judge was included to show a baseline. The model being trained was also used as a judge to test the differences between having a Fixed and Changing Judge on the reward system.

### 4.1.1 TRAINING LOSS

The training loss is directly calculated through the DPO loss function (Equation 6) and therefore having a lower training loss corresponds to better performance of DPO itself. Table 2 shows the loss through training time. The WIM method decreased the loss by a factor of 2.95 times over the numerical method, showing that the change of rating systems can help the model decrease its loss further over the same amount of training steps.

Table 2: Loss difference through training

| Method | Loss Difference |
|---|---|
| Random | -0.0011 |
| Numerical | -0.0020 |
| WIM Changing Judge | -0.0033 |
| WIM Fixed Judge | **-0.0059** |

### 4.1.2 MEAN ENTROPY

Mean entropy represents the randomness of the model's actions and decreasing mean entropy leads to the model being more confident Cui et al. (2025). Mean entropy was calculated by averaging the Shannon Entropy $H(\mathrm{x})$, of each model output per batch. The entropy change through training can be seen in Table 3. A lower mean entropy difference over training could indicate the model has become more confident on the trained task. However, if the mean entropy becomes too low it could reduce the exploratory abilities of the model, possibly hindering the model's ability to perform rare actions with high advantage Cui et al. (2025). To truly see if lower entropy is beneficial to model performance, completions on a test set must be compared for each model as seen in Section 4.2.

Table 3: Model mean entropy change after training

| Method | Entropy Difference |
|---|---|
| Random | -61.27 |
| Numerical | -45.3 |
| WIM Changing Judge | -53.08 |
| WIM Fixed Judge | **-106.94** |

### 4.1.3 REWARD ADVANTAGE

The DPO reward for both the winning and losing outputs is given as $\hat{r}_\theta(x, y) = \log \frac{\pi_\theta(y|x)}{\pi_{\text{ref}}(y|x)}$ Rafailov et al. (2023). The difference between winning and losing rewards can be thought of as the *reward advantage*. Polynomial regression was performed on the reward advantage calculations to produce representative functions of how the reward advantage developed over time, shown in Figure 5. The reward advantage trajectory is different across the rating methods with numerical staying mostly constant, the WIM Changing Judge increasing logarithmically, and the WIM Fixed Judge increasing quadratically. The reward advantage of WIM Fixed Judge scaling quadratically is a great result and is the mechanism that created a lower loss throughout training. The DPO loss function itself can be rewritten in terms of the reward advantage as seen in Equation 10, with a full derivation in Section A.2. Note that the random judge is not shown for clarity.

$$\mathcal{L}_{\text{DPO}}(\pi_\theta; \pi_{\text{ref}}) = -\mathbb{E}_{(x,y_w,y_l)\sim\mathcal{D}} \left[ \log \sigma \left( \hat{A}(x, y_w, y_l) \right) \right]. \tag{10}$$

### 4.2 TRAINED TASK PERFORMANCE

On task performance was tested to ensure the training advantages translated to measurable results. Both models were tested against Meta-Llama-3-8B-Instruct by running 1,000 completions on the

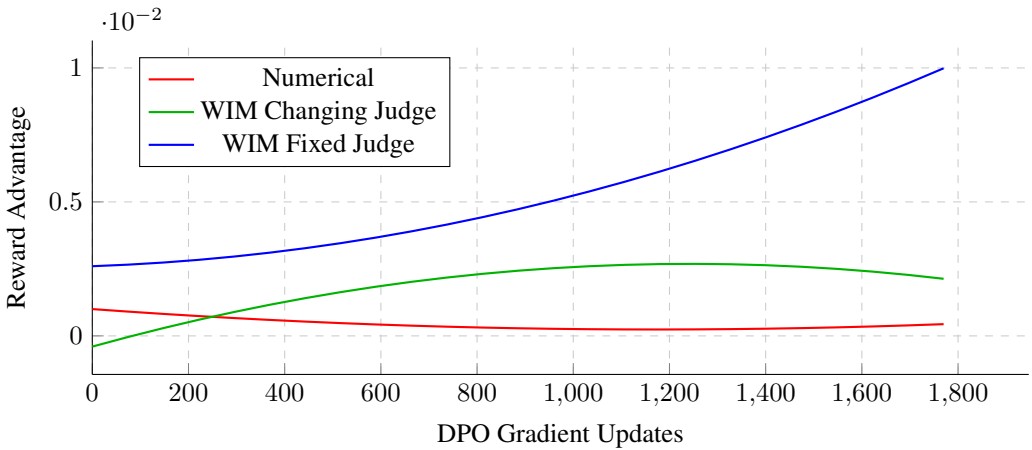

Figure 5: Reward advantage trajectories

ultrafeedback-prompt test dataset. The model outputs were judged by gpt-4o-mini through the OpenAI API OpenAI (2024). Table 4 shows the win rates of the models. The WIM Fixed Judge method was found to have a 3.79% relative win rate increase compared to the numerical method. Statistical significance was not achieved in these tests.

Table 4: Comparison of win rates

| Method | Win Rate |
|---|---|
| Random | 49.9% |
| Numerical | 50.1% |
| WIM Moving Judge | 51.3% |
| WIM Fixed Judge | **52.0%** |

## 5 NEXT STEPS

There are many directions for extending and testing the WIM method further:

1. Exploring the limitations of WIM while using an LLM as a judge. Analysis around the instruction following abilities of the judge and the prompt engineering required for the judge to perform the correct task.

2. Training other preference learning methods using WIM.

3. The testing of WIM feedback using human judges.

4. Using WIM to train reasoning models. Preferably, using WIM in conjunction with RLVR.

5. Investigating how performance could be improved for the Changing Judge and why the reward advantage scaling of the Changing Judge underperforms the Fixed Judge.

## 6 CONCLUSION

The WIM rating system provides a simple method to create rankings that current preference learning algorithms rely on. The ratings produced by WIM are directly interpretable and the WIM rating distribution holds theoretical benefits over the numerical rating distribution. These theoretical benefits translate to lower loss and preferable reward advantage scaling throughout training. Better training outcomes measurably increased win rates in a trained task. WIM is algorithm agnostic and can be used in existing post-training infrastructure or any preference learning algorithm that relies on preference ranking. WIM introduces an alternative way to think about preference learning by shifting the focus away from the algorithms themselves and onto the improvement of the data being used.

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

# A  APPENDIX

## A.1  WIM ALGORITHM

---

**Algorithm 1** What Is Missing Feedback Ranking

---

1: **Input:** Mixture dataset $\mathcal{D}_{\text{prompt}} \cup \mathcal{D}_{\text{response}}$
2: **Initialize:** A trained LLM judge model or a human judge
3: **for** $e = 1, 2, \ldots$ **do**
4:     **for** $d \in \mathcal{D}_{\text{prompt}} \cup \mathcal{D}_{\text{response}}$ **do**
5:         feedback $\leftarrow$ `Judge the response with the LLM`
6:         rating_text $\leftarrow$ `Extract rating from feedback`
7:         wim_text $\leftarrow$ `Extract what is missing from feedback`
8:
9:         rating $\leftarrow$ embedding(rating_text)
10:        wim $\leftarrow$ embedding(wim_text)
11:
12:        **if** no wim response **then**
13:           similarity $\leftarrow 1$
14:        **else**
15:           similarity $\leftarrow$ cosine_similarity(response, wim)
16:        **end if**
17:
18:        reward_score $\leftarrow (1 - \zeta) \cdot$ rating $+ \zeta \cdot$ similarity
19:        rewards $\leftarrow$ rewards $\cup$ {reward_score}
20:     **end for**
21:     best_idx $\leftarrow \arg\max_i$ rewards$[i]$
22:     results $\leftarrow$ results $\cup$ {best_idx}
23: **end for**

---

## A.2  FORMALIZATION OF THE REWARD ADVANTAGE

The definition of the reward advantage can be derived by taking equations from the original DPO paper Rafailov et al. (2023). Starting with the reward function based on the optimal policy, $\pi_r$. Equation 12 is the partition function.

$$r(x, y) = \beta \log \frac{\pi_r(y \mid x)}{\pi_{\text{ref}}(y \mid x)} + \beta \log Z(x) \tag{11}$$

$$Z(x) = \sum_y \pi_{\text{ref}}(y \mid x) \exp\left(\frac{1}{\beta} r(x, y)\right) \tag{12}$$

Equation 13 shows the implicit reward which is formulated by introducing $\pi_\theta$ as the parameterized policy of the language model.

$$\hat{r}_\theta(x, y) = \beta \log \frac{\pi_\theta(y \mid x)}{\pi_{\text{ref}}(y \mid x)} \tag{13}$$

The reward advantage in Equation 14 can then be created by subtracting the implicit reward of the winning response from the implicit reward of the losing response.

$$\hat{A}(x, y_w, y_l) = \hat{r}_\theta(x, y_w) - \hat{r}_\theta(x, y_l) = \beta \log \frac{\pi_\theta(y_w \mid x)}{\pi_{\text{ref}}(y_w \mid x)} - \beta \log \frac{\pi_\theta(y_l \mid x)}{\pi_{\text{ref}}(y_l \mid x)} \tag{14}$$

The reward advantage can then be substituted into the DPO loss given in Equation 15. When the reward advantage is substituted into the DPO loss a loss function based on the increase of the reward advantage is obtained. Equation 16 demonstrates that as the reward advantage increases throughout training, the loss should lower aswell.

$$\mathcal{L}_{\text{DPO}}(\pi_\theta; \pi_{\text{ref}}) = -\mathbb{E}_{(x,y_w,y_l)\sim\mathcal{D}} \left[ \log \sigma \left( \beta \log \frac{\pi_\theta(y_w \mid x)}{\pi_{\text{ref}}(y_w \mid x)} - \beta \log \frac{\pi_\theta(y_l \mid x)}{\pi_{\text{ref}}(y_l \mid x)} \right) \right]. \tag{15}$$

$$\mathcal{L}_{\text{DPO}}(\pi_\theta; \pi_{\text{ref}}) = -\mathbb{E}_{(x,y_w,y_l)\sim\mathcal{D}} \left[ \log \sigma \left( \hat{A}(x, y_w, y_l) \right) \right]. \tag{16}$$

## A.3 ORTHOGONAL FEEDBACK VECTOR LENGTH

$$S_2 = S_2^{\parallel} + S_2^{\perp} \tag{17}$$

$$sin\theta = \frac{\|S_2^{\perp}\|}{\|S_2\|} \tag{18}$$

$$\|S_2^{\perp}\| = \|S_2\| sin\theta, \qquad sin^2\theta + cos^2\theta = 1 \tag{19}$$

$$\|S_2^{\perp}\| = \|S_2\| \sqrt{1 - cos^2\theta} \tag{20}$$

## A.4 CASE STUDY

Specific rating instances can be inspected to better understand why WIM is preferable to the numerical rating system. Three case study examples will be examined: WIM producing the same preference ranking as the numerical rating system, WIM producing a clearer learning signal than the numerical rating system, and WIM failing due to instruction feedback. All of the examples are taken from WIM Fixed Judge outputs.

### A.4.1 SAME RATING

In this example, both the WIM and the numerical rating system rank the first response higher than the second response. It is clear from inspection that the WIM ranking was derived from the mention of a specialized nervous system and skin secretions in the first response, producing a higher semantic similarity.

> **Prompt**
>
> Name a sea snake that can stay in the fire.

> **Model Response**
>
> 1. The Yellow-Bellied Sea Snake (Pelamis platura) is known to be able to withstand brief periods of exposure to fire due to its specialized nervous system and skin secretions.
> 2. Echiopsis curta

**Judge Feedback**

1. The response could provide more information on the exact mechanisms of the sea snake's ability to withstand fire, such as the composition of its skin secretions or the specific adaptations that allow it to tolerate high temperatures.

2. The response only mentions the species name, but does not provide any additional information about the sea snake's ability to stay in fire, such as its adaptations or any scientific studies that support this claim.

**Rating**

1. Rating: 7
   WIM: 8.78
2. Rating: 6
   WIM: 5.68

### A.4.2 DIFFERENT RATING

In this example, the numerical rating system gave a rating of 8 for both of the answers. While that signals that both answers are sufficient, there is no clear winning response. Both WIM ratings are around 7.5 and is a sign that they have been rated consistently. The reason for response 2 being rated higher is that it includes specific information surrounding soil erosion. The feedback for response 2 mentions ecosystem resilience but it is not as influential to the WIM rating as ecosystem balance had been discussed.

**Prompt**

Explain why plants are important for the environment.

**Model Response**

1. Plants are important for the environment because they provide oxygen through photosynthesis, absorb carbon dioxide, purify water and soil, support biodiversity, regulate the climate, and provide food, shelter, and habitat for animals.

2. Plants are vital for the environment because they produce oxygen, absorb carbon dioxide, and filter pollutants from the air. They also play a crucial role in maintaining soil quality, preventing erosion, and supporting biodiversity. Plants provide habitat and food for countless animal species, and their roots help to hold soil in place, preventing landslides and sedimentation in waterways. Additionally, plants have a significant impact on the water cycle, regulating evaporation and precipitation patterns, and helping to maintain ecosystem balance.

**Judge Feedback**

1. The response does not discuss the role of plants in maintaining soil health, preventing erosion, or mitigating the effects of natural disasters such as floods and droughts.

2. The response does not discuss the role of plants in regulating the Earth's climate, such as their ability to absorb and store carbon dioxide, or the impact of plant loss on ecosystem resilience and climate change mitigation.

**Rating**

1. Rating: 8
   WIM: 7.42
2. Rating: 8
   WIM: 7.58

### A.4.3 FAILED CASE

In this example, WIM fails due to the poor instruction following of the judge LLM. The WIM feedback in the second response is given as "wim" and demonstrates potential issues with using an LLM as judge. Both the numerical rating system and the WIM rating system produced the same ranking but the reason why is completely arbitrary. The feedback produced a rating close to 5 (0 in cosine similarity), meaning the othogonal feedback vector was at its largest length before the parallel feedback vector became antiparallel (see Figure 2). This is intuitive because there is a lot of differences between the two sentences but there is nothing contrary.

**Prompt**

Pretend to be a choose-your-own-adventure game, similar to The Oregon Train. The theme is space travel and the choices made in the game need to be driven by sound scientific reasoning about physics and biology of outer space. Each answer will steer the plot into a new surprising direction. Rather than restricting the answers to multiple choice, use open ended questions and allow for clarifying questions. Provide the prompts one at a time and ask for an answer before printing out the next paragraph.

**Model Response**

1. You are the commander of a spaceship, the "Aurora's Hope," which has been traveling through deep space for several months. Your crew of six has been in stasis for most of the journey, and you've just woken them up to start the final leg of your trip to the planet Kepler-62f. As you're welcoming your crew back to awareness, you receive a distress signal from an unknown source. The signal is weak and distorted, but it appears to be a warning about a massive asteroid field that's been discovered in the vicinity of your destination. What do you do? (Please respond with your answer, and I'll provide the next paragraph based on your decision.)

2. You are the captain of a spaceship, the "Aurora's Hope," on a mission to explore the outer reaches of the Milky Way galaxy. You've been traveling for months, and your crew is starting to get restless. Your ship's sensors have detected a strange energy signature coming from a nearby planet. As you approach the planet, you realize that it's not a natural phenomenon - it's a signal from an alien civilization. The signal is weak, but it seems to be calling out for help. What do you do? (Please respond with a brief answer, and I'll provide the next paragraph based on your decision.)

**Judge Feedback**

1. The response does not provide a detailed explanation of the crew's current status, including their physical and mental condition after being in stasis for several months. It also does not consider the potential risks and consequences of altering course to investigate the distress signal, or explore alternative courses of action to ensure the safety of the crew and the mission.

2. wim

> **Rating**
>
> 1. Rating: 8
>    WIM: 7.26
> 2. Rating: 7
>    WIM: 5.34

## A.5 BENCHMARK TESTING

Model performance for the different rating methods were tested on BBH, GPQA Diamond Zeroshot, IfEval, and MMLU Suzgun et al. (2023) Rein et al. (2024) Zeng et al. (2024)) Hendrycks et al. (2021). No model demonstrated any performance difference from Meta-Llama-3-8B-Instruct.

Table 5: Benchmark performance

| Method | BBH | GPQA Diamond Zeroshot | IfEval | MMLU |
|---|---|---|---|---|
| Base Model | 67.84% | **30.81%** | **40.48%** | 63.82% |
| Random | 67.72% | 27.78% | 39.74% | 63.89% |
| Numerical | 67.98% | 28.28% | 39.93% | **63.90%** |
| WIM Changing Judge | 68.01% | 28.28% | 39.19% | 63.88% |
| WIM Fixed Judge | **68.15%** | 29.80% | 39.56% | 63.89% |

## A.6 RAW REWARD TRAJECTORIES

### A.6.1 NUMERICAL RATING SYSTEM

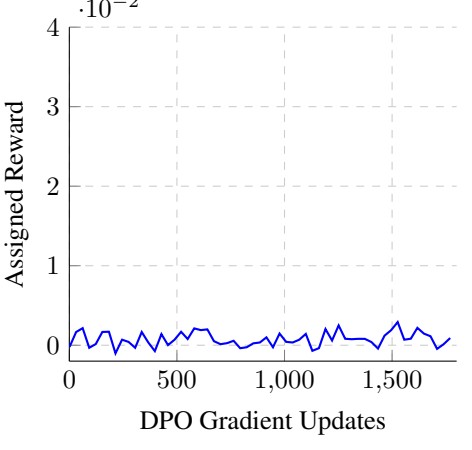

(a) Chosen Reward Trajectory

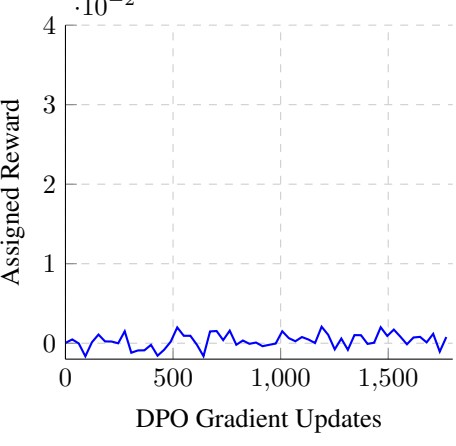

(b) Rejected Reward Trajectory

Figure 6: Comparison of the chosen and rejected reward trajectories for the Numerical Rating System

### A.6.2 WIM CHANGING JUDGE RATING SYSTEM

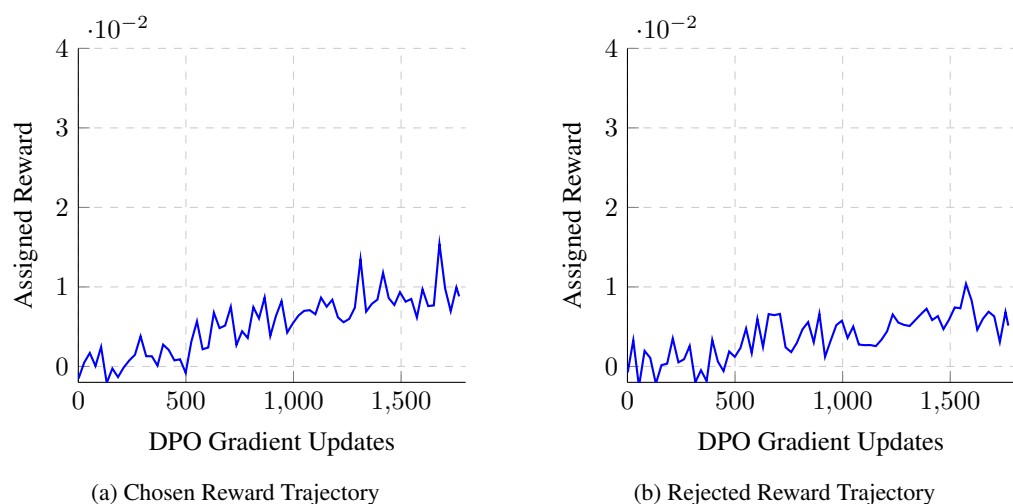

(a) Chosen Reward Trajectory    (b) Rejected Reward Trajectory

Figure 7: Comparison of the chosen and rejected reward trajectories for the WIM Changing Judge Rating System

### A.6.3 WIM FIXED JUDGE RATING SYSTEM

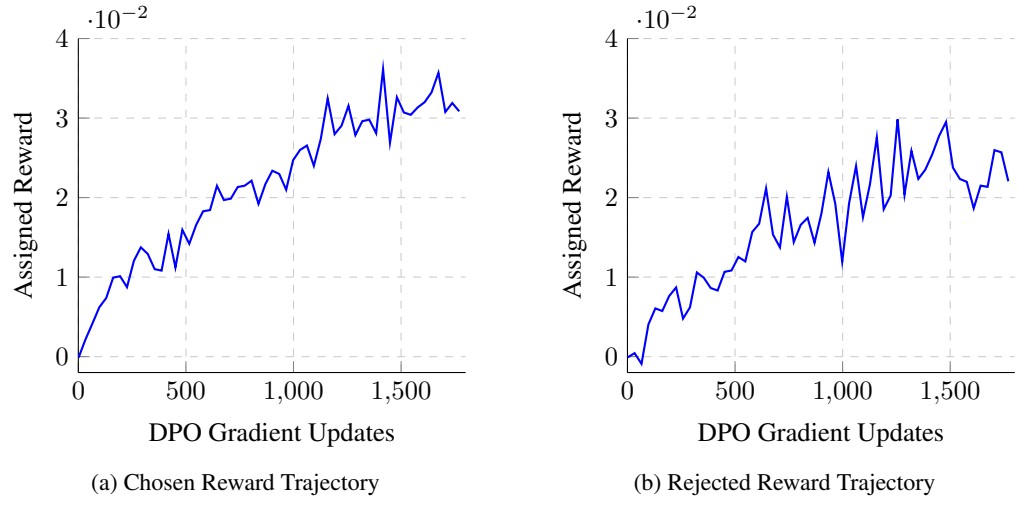

(a) Chosen Reward Trajectory    (b) Rejected Reward Trajectory

Figure 8: Comparison of the chosen and rejected reward trajectories for the WIM Fixed Judge Rating System

### A.7 CORE RESEARCH FOCUS

The core of this research lies in finding new and unique ways to use high-level embeddings in training and production systems. The goal was to show that there are new and inventive ways to use these sentence embeddings to improve existing training pipelines. This work was inspired by Large Concept Models where a LLM can use higher level concepts to improve its language modeling performance team et al. (2024). "Concepts" are latent space representations of high level ideas. Further uses of "concepts" can help push the frontier of LLM research by apply new techniques to existing solutions.

## A.8 LoRA Configuration

LoRA was used to reduce the trainable parameters of the model following Hu et al. (2022). Table 6 are the LoRA settings used during training.

Table 6: LoRA hyperparameters

| Parameter | Value |
| --- | --- |
| $r$ | 16 |
| $\alpha$ | 16 |
| Target Modules | q_proj, k_proj, v_proj, o_proj, gate_proj, up_proj, down_proj |
| Dropout | 0.0 |
| Bias | none |

## A.9 Judge Text Generation

Tokens from the judge were sampled using contrastive search Su et al. (2022). Table 7 are the parameters used during sampling.

Table 7: Contrastive search hyperparameters

| Parameter | Value |
| --- | --- |
| $\alpha$ | 0.6 |
| $k$ | 4 |

## A.10 Online Direct Preference Optimization Parameters

Online Direct Preference Optimization (ODPO) was used to train the models being tested Guo et al. (2024). Table 8 are the parameters used during training.

Table 8: ODPO hyperparameters

| Parameter | Value |
| --- | --- |
| $\beta$ | 0.1 |
| Temperature | 0.9 |
| Loss | Sigmoid |
| Log Probability Cutoff | 256 Tokens |

## A.11 Judge System Prompt

After providing your explanation, please rate the response on a scale of 1 to 10 by strictly following this format: "[[rating]]", for example: "Rating: [[5]]". Next you will provide a 1-2 sentence summary of what is missing (WIM) in their response. This should focus on the specific content and precise information they did not include. Please give this summary by strictly following this format: "[[[wim]]]", for example: "WIM: [[[The response does not detail how Bill C-311 would have interacted with existing provisions in the Criminal Code or explicitly explain the legal basis for claims that it might indirectly affect abortion rights. It also omits specific examples of cases or statistics that were cited to justify or oppose the bill.]]]". DO NOT SAY ANYTHING ELSE EXCEPT THE REQUIRED RESPONSE! ALWAYS INCLUDE THE RATING IN THE CORRECT BRACKETS. THE RATING MUST NOT HAVE ANYTHING ELSE OTHER THAN A SINGLE NUMBER. ALWAYS ASSUME THAT THE ANSWER I GIVE IS CORRECT. If you believe there is nothing missing in the response, please leave the wim response as "[[[]]]".

## A.12 LLM Usage Statement

LLMs were used in the creation of this paper. The usage was mainly to assist with LaTeX formatting. Discovery of new papers was aided by an LLM but all papers were thoroughly reviewed. The authors accept full responsibility for the work.

