# OpenReview forum: "What Is Missing: Interpretable Ratings for Large Language Model Outputs"
_ICLR.cc/2026/Conference — Submitted to ICLR 2026_

### Official Review · Reviewer_c1ep · 2025-10-29

**Soundness:** 1
**Presentation:** 2
**Contribution:** 2
**Rating:** 2
**Confidence:** 5

**Summary:**

This paper introduces the What Is Missing (WIM) rating system as an improvement over traditional preference learning methods for LLMs. Current methods like PPO and DPO rely on subjective direct rankings or numerical ratings (e.g., 1-10 scales) to learn human preferences. The authors argue that these approaches are problematic because a single numerical rating poorly captures the complexity of human language, ratings lack interpretability, and discrete rating systems frequently produce duplicate scores that prevent generating meaningful learning signals. The WIM system addresses these shortcomings while remaining compatible with existing training infrastructure.

**Strengths:**

- 1. Ratings are directly derived from human-readable natural language feedback explaining what is missing. Unlike opaque numerical scores, anyone can understand exactly why a rating was assigned, which enables easy debugging and identification of model weaknesses or rating errors.

- 2. WIM produces a continuous-like distribution instead of clustered discrete values, reducing duplicate ratings from 42.78% (numerical systems) to only 2.00%. This generates 47.82% higher average rating deltas between outputs.

- 3. WIM is algorithm-agnostic and works with any preference learning method (PPO, DPO, GRPO, etc.) without modifications. It can be plugged into existing training pipelines saving engineering time and costs, supports multiple judge configurations (human, separate LLM, or self-judging), and can be mixed with other rating methods through a tunable parameter.

**Weaknesses:**

- The authors propose the concept of "interpretable rating" but fail to provide a clear definition of this notion. The interpretability of language models is a highly complex concept that demands a well-defined framework—specifying which type of interpretability is being pursued, e.g., mechanistic explanation as in mechanistic interpretability [1] or a theoretical perspective from training dynamics [2].

- Rating alone means nothing for interpretability because LLMs can still exhibit behaviors like sycophancy [3] or deception [4] to generate unintended responses (which also occurs with benign inputs [5]). For example, they may give relatively positive responses for a bad example when evaluating certain inputs, and thus semantic cosine similarity cannot serve as a good metric.

- Much more literature review is needed. In terms of improving the accuracy and performance of LLM-as-a-judge, there are many alternative solutions such as natural language as reward [6] and improved versions of LLM-as-a-judge (please refer to https://github.com/llm-as-a-judge/Awesome-LLM-as-a-judge for more details). As for algorithmic improvements of DPO, there are also many variants that align LLMs with further alignment from representation engineering or feature level (e.g., FPO).  There are almost no baselines comparing against these two fields, which severely undermines the paper's contribution.

- The experiments cannot adequately support the results. Only LLaMA-3 is used, and although I understand that research papers with limited resources should not be required to conduct excessive experiments, relying on only one model with one parameter size cannot demonstrate effectiveness across different model architectures (Qwen performs really differently from LLaMA in tasks like reasoning) and scaling (will this work for larger models?).

[1] https://transformer-circuits.pub/2025/attribution-graphs/biology.html

[2] https://arxiv.org/html/2505.17646v1

[3] https://arxiv.org/abs/2310.13548

[4] https://arxiv.org/abs/2501.16513

[5] https://arxiv.org/abs/2508.06361

[6] https://www.arxiv.org/abs/2506.03637

**Questions:**

1. Can the authors explain what kind of interpretability they have achieved in this paper?

2. There is no explicit performance gain in benchmarks like MMLU. Are there any reasons for this?

3. Please provide more baselines, e.g., natural-language-as-reward [6] and other improved LLM-as-a-judge methods.

4. Have the authors tested this method on other model architectures (e.g., Qwen, Mistral) to verify generalizability?

5. What are the individual contributions of each component in your proposed method? Please provide ablation studies to demonstrate which parts are essential for the claimed improvements.

---

### Official Review · Reviewer_WnUb · 2025-10-31

**Soundness:** 1
**Presentation:** 3
**Contribution:** 2
**Rating:** 2
**Confidence:** 4

**Summary:**

In this paper, the authors introduce "What is Missing (WIM)", an approach to rate responses of an LLM and use this ratings for preference optimization.
The method employs a Judge (LLM or human) to produce what is missing from a candidate response. Then the cosine similarity is calculated between the candidate response and the Judge's output, and finally the score is mapped to a 1-10 (continuous) rating scale.
The main advantages of the method are the interpretable nature of the ratings --the Judge's output can be directly inspected-- and the continuous (rather than discrete) nature of the rating score, allowing the method to model potentially more expressive ratings.
The main idea is clearly conveyed and some preliminary experiments are presented, showing the promise of the method. However, many of the arguments and claims made throughout the paper need further rigorous experimentation.

**Strengths:**

S1. The framework proposed is flexible enough to be adapted to a plethora of preference optimization (PO) methods, with the possibility to use either human or LLM-based judges.
S2. The method tackles the lack of interpretability and expressiveness in preference ratings, a well-known limitation in PO methods.

**Weaknesses:**

The paper mainly lacks rigorous experiments to support the claimed benefits of the model, such as how it compares to strong, simpler baselines (PPO, DPO) in well-stablished experimental setups for preference optimization (e.g. HH, TL;DR, AlpacaEval2, among many others)
As such, it is difficult to make grounded conclusions about the contributions of this paper.

**Questions:**

- One of the main assumptions the paper makes is that the cosine similarity of sentence embeddings can be used as an indicator of semantic overlap between two responses. However, this assumption needs to be confirmed experimentally for each use case. Previous work has found that contextual embeddings of tokens and sentences might not be appropriate for semantic similarity comparisons, since a few dimensions might dominate the representation, leading to misleading or inflated cosine similarities [1,2].
Please consider doing a preliminary, sanity-check experiment to confirm that this is not the case, if you haven't already done so.
- Could you please elaborate why an online optimization algorithm is more appropriate for your methodology rather than an offline one?

The experimental setup could greatly benefit from the following
- Direct comparison against other candidate ranking or selection strategies, using basic PO methods such as PPO, DPO, and compared over standard PO benchmarks, e.g. AlpacaEval2.
- Usage of available PO datasets with ratings, e.g. Argilla DPO Mix 7K
- Training model on full precision, even models as small as 1B
- Experiments over a range of preference optimization scenarios well investigated in previous work, such as helpfulness and harmfulness, summary quality control (TL;DR), toxicity, etc. See [3,4].
- An analysis on the influence of the difference in length between a response and the Judge output. Ideally, WIM will not show significant bias against extremely short or extremely long responses.

- Table 1 could be explained directly in the text body and save space.
- Claims and sections that would need experimental evidence:
  - Section 3.3, "WIM could improve training of reward models"
  - L352, "WIM can be integrated into existing training pipelines"... currently tested only for online DPO. Integration on other PO methods would give us a better picture.


[1] https://aclanthology.org/2021.emnlp-main.372.pdf
[2] https://aclanthology.org/2020.emnlp-main.733/
[3] https://arxiv.org/abs/2305.18290
[4] https://aclanthology.org/2024.findings-acl.592.pdf

---

### Official Review · Reviewer_x7hE · 2025-11-01

**Soundness:** 1
**Presentation:** 1
**Contribution:** 1
**Rating:** 0
**Confidence:** 4

**Summary:**

This paper is motivated by the ambiguity and subjectivity of human preference over model outputs, specifically as expressed by boolean or continuous rankings. It instead proposes an automatically inferred ranking, wherein a judge (human or LLM) provides feedback on which information is missing from model outputs, and the ranking is inferred through the cosine similarity between feedback and the original outputs.

The paper is unfortunately very difficult to follow and the efficacy of its proposed method is not well supported. The theoretical advantages are not clearly demonstrated, and in experiments the proposed method is not significantly better than a randomized preference metric.

**Strengths:**

- The paper is well motivated, in that an automatic method to determine preferences would be beneficial.

**Weaknesses:**

- The premise of the paper is counter-intuitive. The proposed metric seems to be limited to instances were preference can be determined through key missing information. However, in those cases the human provided preferences should not be ambiguous nor subjective. In contrast, in cases where there is low human agreement, e.g. preference over stylistic choices in language, there would be no missing information for this metric to capture.
- The qualitative analysis is vague and based on undisclosed human-provided rankings. The paper additionally provides no examples of the rankings or the generated feedback.
- The experimental results clearly show that the proposed method is not significantly different from a random baseline. The same is shown for the numerical rankings, which casts doubt on the experimental procedure and/or the rankings themselves. The modest improvements in training loss or reward advantage by themselves cannot support the efficacy of the method.
- The paper could benefit from intensive proofreading to improve its presentation, including confusing claims, citations used in the wrong format, and numerous typos.

**Questions:**

- Please elaborate on how the numerical rankings were obtained in Section 3. Provide information on who the annotators were and their relation to the paper. Were the annotators familiar with the task and aware of the missing information in each output?

---

### Official Review · Reviewer_txzA · 2025-11-03

**Soundness:** 2
**Presentation:** 2
**Contribution:** 3
**Rating:** 2
**Confidence:** 3

**Summary:**

This paper proposes a novel way to provide a score to a summary written by an LLM. In particular, it asks a judge LLM to provide what is missing in the summary and take the cosine similarity between the summary and the response from the judge LLM. The authors provide evidence that support their claim that their WIM score improves the task performance.

**Strengths:**

The strength is the novelty of the idea that tries to set up an adversary-like LLM.

**Weaknesses:**

Overall, I feel that the paper has a nice idea but the evidence is on the weak side and the evaluation is far from comprehensive.

- There is a frequent use of passive voice, which makes it difficult to immediately catch if the subject is the authors or existing literature or someone else. There are excellent web articles about why passive voice can make the writing less clear. Please consider switching to active voice to improve readability.
- Figure 3 does not provide a fair comparison. The problem is that the frequency of WIM rating looks smaller because the binning is thinner. Please plot it again with 10 bins for the WIM rating. The frequency will surely increase.
- Section 3 could've been more comprehensive by showing that the same kind of results are reproduced across various models.
- Section 4.1.1 Training loss: I am really not convinced that this is providing a meaningful signal because here, the loss function is using a different reward model. It depends on beta. Ideally, for each method, we could tune beta and report the best. I did not really catch the point trying to make here.
- Section 4.1.3 reward advantage: I am not sure what signals we are catching here. Large reward advantage just means that we are going away from piref. Simply using a larger step size would achieve this, so it is not clear what the plot signals.
	- Also, the authors say "WIM changing judge increasing logarithmically", but the curve looks like an upside-down U shape. Why do they think it is a logarithmic curve? Also, the right way to verify if it is logarithmic is to make the x-axis logarithmic and verify if it forms a line..
	- Also, it is unclear how much the fitting was accurate. How do we know if the same trend will persist when run with a different random seed? I would love to see the original data (training dynamics) used for fitting the polynomial curve.
- Overall, there is no explanation on why WIM changing judge vs fixed judge makes a difference


minor comments
- L218: in what specific sense does it help interpretability?
- I was not able to find the definition of $\hat r$ in Eq (7), but maybe I could have missed it. I found it later in L415, but it was not clear to me that it was exactly that one. I think we would have to multiply $\beta$ to the definition of $\hat r$ therein.
- is moving judge = changing judge?

**Questions:**

See the weakness above.

---

### Meta-Review · Area_Chair_sNT4 · 2025-12-16

**Summary:**

This paper proposes the "What Is Missing" (WIM) rating system for preference learning in LLMs, where natural language feedback is converted to ratings via cosine similarity of sentence embeddings. While reviewers acknowledged the novelty of the idea and its potential interpretability benefits, all four reviewers assigned reject ratings, citing fundamental weaknesses that prevent acceptance. There is no author response to address these concerns. I recommend rejection.

**Reviewer Scores:**

N/A

---

### Decision · Program_Chairs · 2026-01-26

Reject